# Semantic Segmentation of Urinary Bladder Cancer Masses from CT Images: A Transfer Learning Approach

**DOI:** 10.3390/biology10111134

**Published:** 2021-11-04

**Authors:** Sandi Baressi Šegota, Ivan Lorencin, Klara Smolić, Nikola Anđelić, Dean Markić, Vedran Mrzljak, Daniel Štifanić, Jelena Musulin, Josip Španjol, Zlatan Car

**Affiliations:** 1Faculty of Engineering, University of Rijeka, Vukovarska 58, 51000 Rijeka, Croatia; sbaressisegota@riteh.hr (S.B.Š.); ilorencin@riteh.hr (I.L.); nandelic@riteh.hr (N.A.); dstifanic@riteh.hr (D.Š.); jmusulin@riteh.hr (J.M.); car@riteh.hr (Z.C.); 2Clinical Hospital Center Rijeka, Krešimirova 42, 51000 Rijeka, Croatia; klara.smolic@gmail.com (K.S.); dean.markic@medri.uniri.hr (D.M.); josip.spanjol@medri.uniri.hr (J.Š.); 3Faculty of Medicine, Branchetta 20/1, University of Rijeka, 51000 Rijeka, Croatia

**Keywords:** artificial intelligence, computer tomography, machine learning, semantic segmentation, urinary bladder cancer

## Abstract

**Simple Summary:**

Bladder cancer is a common cancer of the urinary tract, characterized by high metastatic potential and recurrence. The research applies a transfer learning approach on CT images (frontal, axial, and saggital axes) for the purpose of semantic segmentation of areas affected by bladder cancer. A system consisting of AlexNet network for plane recognition, using transfer learning-based U-net networks for the segmentation task. Achieved results show that the proposed system has a high performance, suggesting possible use in clinical practice.

**Abstract:**

Urinary bladder cancer is one of the most common cancers of the urinary tract. This cancer is characterized by its high metastatic potential and recurrence rate. Due to the high metastatic potential and recurrence rate, correct and timely diagnosis is crucial for successful treatment and care. With the aim of increasing diagnosis accuracy, artificial intelligence algorithms are introduced to clinical decision making and diagnostics. One of the standard procedures for bladder cancer diagnosis is computer tomography (CT) scanning. In this research, a transfer learning approach to the semantic segmentation of urinary bladder cancer masses from CT images is presented. The initial data set is divided into three sub-sets according to image planes: frontal (4413 images), axial (4993 images), and sagittal (996 images). First, AlexNet is utilized for the design of a plane recognition system, and it achieved high classification and generalization performances with an $\bar{AUC_{micro}}$ of 0.9999 and $\sigma (AUC_{micro})$ of 0.0006. Furthermore, by applying the transfer learning approach, significant improvements in both semantic segmentation and generalization performances were achieved. For the case of the frontal plane, the highest performances were achieved if pre-trained ResNet101 architecture was used as a backbone for U-net with $\bar{DSC}$ up to 0.9587 and σ(DSC) of 0.0059. When U-net was used for the semantic segmentation of urinary bladder cancer masses from images in the axial plane, the best results were achieved if pre-trained ResNet50 was used as a backbone, with a $\bar{DSC}$ up to 0.9372 and σ(DSC) of 0.0147. Finally, in the case of images in the sagittal plane, the highest results were achieved with VGG-16 as a backbone. In this case, $\bar{DSC}$ values up to 0.9660 with a σ(DSC) of 0.0486 were achieved. From the listed results, the proposed semantic segmentation system worked with high performance both from the semantic segmentation and generalization standpoints. The presented results indicate that there is the possibility for the utilization of the semantic segmentation system in clinical practice.

## 1. Introduction

Urinary bladder cancer is one of the ten most common cancers worldwide. It is characterized by an cancerous alteration and uncontrollable growth of bladder tissue, typically urothelial cells, which develop into a tumor and can spread into other organs. Patients who suffer from bladder cancer may exhibit various symptoms, such as painful and frequent urination, blood in the urine, and lower back pain. Research indicates that tobacco smoking largely increases the risk of developing bladder cancer [1]. Other external factors that may increase the risk of bladder cancer are a previous exposure to radiation, frequent bladder infections, obesity, and exposure to certain chemicals, such as aromatic amines [2,3].

Multiple different pathohistological subtypes of bladder cancer exist, including urothelial carcinoma (transitional cell carcinoma)—the most common type of bladder cancer [4]; squamous cell carcinoma—which is rare and associated with chronic irritation of the bladder commonly due to infections or prolonged catheterization [5]; adenocarcinoma—a very rare subtype of cancer, arising in other, neighboring organs as well [6]; small cell carcinoma—a highly aggressive type of cancer with a high metastatic potential, commonly diagnosed at advanced stages [7]; and sarcoma—an extremely rare and aggressive type of bladder cancer [8].

Diagnosis of bladder cancer is commonly performed using cystoscopy, a procedure in which a fiber-optic instrument is passed through the urethra into the bladder and an optical evaluation is performed by a specialist in vivo [9]. The process is significantly less invasive than a biopsy but also has a lower success rate in certain cases—such as distinguishing carcinoma in-situ from scarring or inflammatory changes [10]. As this procedure utilizes a digital camera, previous work [11,12] has shown the ability to improve the results of the procedure through the application of Artificial Intelligence (AI) machine learning (ML) algorithms. This indicates that AI methods could be applied in connected, similar diagnostic problems.

Computed tomography (CT) scans are a commonly used medical diagnostic imaging method in which multiple X-ray measurements are taken to produce tomographic images of a patient’s body, allowing the examination of patient’s organs without the need for a more invasive procedure [13]. Today, multi-detector CT scanners, with 64 to 320 rows of detectors, are used combined with helical image acquisition techniques, as they minimize the exposure to the radiation and can generate sagittal, axial, and frontal images of the body during a single breath-hold [14]. CT urography and CT of the abdomen and pelvis are contrast-enhanced imaging methods for the detection and staging of bladder cancer that are able to differentiate healthy from cancer-affected regions of the bladder [15,16].

Non-ionic monomer iodinated contrast agents are administered intravenously for arterial opacification and parenchymal enhancement, which helps with better delineation of soft tissue [17]. Acquired images are regularly inspected and interpreted by the radiologist, who provides detailed descriptions of the urinary bladder [18]. In post-processing, the radiologist can mark and measure the suspected tumor or create a 3D recreation of a urinary system. Detection of urinary bladder cancer by using CT urography has shown high performance, and it can be concluded that CT urography can be used alongside cystoscopy for detecting urinary bladder cancer [19].

Varghese et al. (2018) [20] demonstrated the application of semi-supervised learning through denoising autoencoders to detect and segment brain lesions. A limited number of patients was used (20, 40, and 65), but despite this, the method displayed good performance on low-grade glaucoma (LGG) segmentation. Ouyang et al. (2020) [21] demonstrated a self-supervised approach to medical image segmentation. The researchers applied a novel few-shot semantic segmentation (FSS) framework for general applicability that achieved good performance in three different tasks: abdominal organ segmentation on images collected through both CT and MRI procedures and cardiac segmentation for MRI images.

Renard et al. (2020) [22] showed the importance of segmentation using deep learning algorithms and proposed three recommendations for addressing possible issues, which are an adequate description of the utilized framework, a suitable analysis of various sources, and an efficient evaluation system for the results achieved with the deep learning segmentation algorithm. Zhang et al. (2020) [23] discussed the applications of deep-stacked data—an application of multiple stacked image transformations and their influence on the segmentation performance with tests performed on multiple three-dimensional segmentation tasks—the prostate gland, left atrial, and left ventricle.

Zhang et al. (2020) [24] integrated an Inception-ResNet module and U-Net architecture through a dense-inception block for feature extraction. The proposed model was used for the segmentation of blood vessels from retina images, lung segmentation of CT data, and an MRI scan of the brain for tumor segmentation, on all of which, it achieved extremely high dice scores.

Liu et al. (2020) [25] demonstrated the application of segmentation for the diagnosis of breast cancer, focusing on the application of Laplacian and Gaussian filters on mammography images available in the MIAS database. The performance was compared to different filters, such as Prewitt, LoG, and Canny, with the tested solutions providing comparable or better performance. Wang et al. (2020) [26] also demonstrated the application of image segmentation on breast cancer nuclei. The researchers applied the U-Net++ architecture, with Inception-ResNet-V2 used as a backbone, allowing for increased performance compared to previous research.

Hongtao et al. (2020) [27] demonstrated the application of segmentation and modeling of lung cancer using 3D renderings created from CT images. The segmentation performed using MIMICS17.0 software and demonstrated high precision; however, due to software limitations, the exact coordinates of tumor location cannot yet be exported. Yin et al. (2020) [28] demonstrated the application of a novel medical image segmentation algorithm—balanced iterative reducing and clustering using hierarchies (BIRCH). The method was applied to brain cancer imagery with the experimental results demonstrating that segmentation accuracy and speed can be applied through the BIRCH application.

Qin et al. (2020) [29] proposed a novel Match Feature U-Net, a symmetric encoder. The method is compared to U-Net, U-net++, and CE-Net showing improvements in multiple image segmentation tasks: nuclei segmentation in microscopy images, breast cancer cell segmentation, gland segmentation in colon histology images, and disc/cup segmentation. Li et al. (2020) [30] demonstrated edge detection through image segmentation algorithms on the three dimensional image reconstruction. The proposed method achieved accuracy above 0.95 when applied with a deep learning algorithm.

Kaushal et al. (2020) [31] showed the application of an algorithm based on the so-called Firefly optimization, with the application on breast cancer images. The proposed method was capable of segmenting images despite their type and modality with effectiveness comparable to or exceeding other state-of-the-art techniques. Alom et al. (2020) [32] displayed the application of improved deep convolutional networks (DCNN) on skin cancer segmentation and classification. The authors proposed NABLA-N, a novel network architecture that achieved an accuracy of 0.87 on the ISIC2018 dermoscopic skin cancer data set.

Li et al. (2020) [33] demonstrated the application of a nested attention-aware U-Net on CT images for the goal of liver segmentation. The authors concluded that the proposed novel method achieved competitive performances on the MICCAI 2017 Liver Tumor Segmentation (LiTS) Challenge Dataset. Tiwari et al. (2020) [34] displayed the application of the fuzzy inference system. The authors applied a pipeline consisting of preprocessing, image segmentation, feature extraction, and the application of fuzzy inference rules, which are capable of identifying lung cancer cells with high accuracy.

Monteiro et al. (2020) [35] demonstrated the use of CNNs for multiclass semantic segmentation and the quantification of traumatic brain injury lesions on head CT images. The patient data was collected in the period between 2014 and 2017, on which the CNN was trained for the task of voxel-level multiclass segmentation/classification. The authors found that such an approach demonstrated a high quality volumetric lesion estimate and may potentially be applied for personalized treatment strategies and clinical research.

Anthimopoulos et al. (2018) [36] demonstrated the use of Dilated Fully Convolutional Networks for the task of semantic segmentation on pathological lung tissue. The authors used 172 sparsely annotated CT scans within a cross-validation training scheme with training done in a semi-supervised mode using labeled and unlabeled image regions. The results showed that the proposed methodology achieved significant performance improvement in comparison to previous research in the field.

Another study regarding segmentation on lung CTs was performed by Meraj et al. (2021) [37] with the goal of performing lung nodule detection. The authors used a publicly available dataset, the Lung Image Database Consortium, upon which filtering and noise removal were applied. The authors used adaptive thresholding and semantic segmentation for unhealthy lung nodule detection, with feature extraction performed via principal component extraction. Such an approach showed results of 99.23% accuracy when the logit boost classifier was applied.

Koitka et al. (2021) proposed an automatic manner of body composition analysis, with the goal of application during routine CT scanning. The authors utilized 3D semantic segmentation CNNs, applying them on a dataset consisting of 50 CTs annotated on every fifth axial slice split into an 80:20 ratio. The authors achieved high results with the average dice scores reaching 0.9553, indicating a successful application of CNNs for the purpose of body composition determination.

To increase the performances of algorithms for the semantic segmentation, we introduce a process of transfer learning. It is important to notice that alongside the performance from the semantic segmentation standpoint, the performance from the generalization standpoint must be evaluated as well. For these reasons, the following questions can be asked:Is it possible to design a semantic segmentation system separately for each plane?Is there a possibility to design an automated system for plane recognition?How does the transfer learning paradigm affect the semantic segmentation and generalization performance of designed U-nets?Which pre-trained architectures achieve the highest performances if used as a backbone for U-net?

To summarize the novelty of the article, the idea is to utilized multi-objective criteria to evaluate the performances of a transfer learning-based system for the semantic segmentation of urinary bladder cancer masses from CT images. The images are captured in three planes: frontal, axial, and sagittal, and the aim of the research is to maximize semantic segmentation performances by dividing data set according to planes and to introduce the system for automatic plane recognition.

At the beginning of the paper, a brief description of the diagnostic procedure is provided together with the problem description. After the problem description, the used data set is presented, followed by a description of the used algorithms. After algorithm description, a mathematical approach to the transfer learning paradigm is presented together with used backbone architectures. In the end, the research methodology is presented followed by the results and discussion.

## 2. Problem Description

With aim of developing an automated system that could be used in the diagnosis and decision making regarding urinary bladder cancer, a system for the semantic segmentation of urinary bladder cancer from CT images is proposed. The proposed approach is based on the utilization of CNN models that are executed on an HPC workstation. The idea behind the utilization of CNN-based semantic segmentation is to use a data set of images with known annotation to train CNN that will later be used to automatically annotate and evaluate new images. As input data to the semantic segmentation system, images of lower abdomen collected with CT are used.

An input image, captured in three planes. First, the classification is performed, in order to determine the plane in which the image was captured. After the plane is determined, the image is used as an input to U-net architecture for that particular plane. Trained U-net architecture is used to create the output mask. The output mask represents the region of an image where urinary bladder cancer is present. Such an output enables automated evaluation of urinary bladder that results in an annotated image that is used to determine the urinary bladder cancer spread. A graphical overview of such a process is presented in Figure 1.

The dataset used in this research was created by using CT images collected in the Clinical Hospital Center of Rijeka, and it consists of CT images of the lower abdomen in three planes:Frontal plane.Sagittal plane.Axial plane.

All images contained in the data set are images where a form of bladder cancer is confirmed. The CT images with confirmed bladder cancer are presented in Figure 2, where Figure 2a represents a CT image in the frontal plane, Figure 2b represents a CT image in the sagittal plane, and Figure 2c represents a CT image in axial plane.

The distribution of the training, validation, and the testing data sets is presented for all three planes in Table 1.

When the numbers of images in each plane are compared, it can be seen that a significantly lower number of images were captured for the case of the sagittal plane, in comparison with the frontal and axial plane. Such an imbalance is a consequence of the fact that the CT urography procedures are, in the case of this research, dominantly performed in the frontal and axial planes only. Furthermore, images captured in the sagittal plane are captured with lower density, resulting in a lower number of images per patient.

The use of this particular data set was approved by Clinical Hospital Center Rijeka, Ethics Board (Kresimirova 42, 51000 Rijeka); under the number 2170-29-01/1-19-2, on 14 March 2019. In order of creating output data for CNN training, image annotation was performed. Such an approach was utilized for creating output masks that represent the bladder region where a malignant mass is present. The annotation was performed by a specialist urologist according to the obtained medical findings.

It is important to emphasize that all images and corresponding medical findings used during this research are validated with additional medical procedures, such as cystoscopy. Medical findings are evaluated by three independent raters—urologists with the experience in the field of radiography, including CT. As a observer agreement measure, Fless’ kappa (κ) coefficient is used [38]. For the case of this study, κ of 0.83 was achieved. This results suggests the conclusion that the agreement of observers is, in this case, of a high degree.

An example of an image annotation procedure is presented in Figure 3, where Figure 3a, Figure 3a,b represent the frontal, sagittal, and axial plane, respectively.

The red areas presented in Figure 3 are used in the creation of output annotation maps that are used during U-net model development [39].

## 3. Algorithm Description

Malignant masses visible from CT images of a urinary bladder can be detected by using a semantic segmentation approach. Cancer detection using a semantic segmentation approach is used to differentiate malignant masses from the remaining part of the urinary bladder and other organs of the lower abdomen. Semantic segmentation is based on the utilization of U-net. Such an approach represents a standard approach in medical image segmentation tasks [40,41], and it is based on generating output masks that represent the area where malignant mass is present [42].

U-net is characterized by its fully convolutional architecture. Such an architecture, in difference with standard CNN architecture, consists only of convolutional layers that are distributed into a contractive and expansive part. The contractive part of U-net is a standard down-sampling procedure similar to every CNN architecture with its convolutional and pooling layers [43]. On the other hand, during the expansive part, an up-sampling procedure is performed [44].

An up-sampled feature map was concatenated with the cropped part of the feature map from the contractive part [45]. The cropping procedure is performed due to the loss of border pixels in contractive of U-net. This procedure is repeated in the order of constructing a segmentation map on the U-net output. The aforementioned semantic segmentation map represents the area of an image where a malignant mass is present and it is, in fact, an output of a semantic segmentation algorithm. The described approach is presented with a block scheme in Figure 4.

The CT procedure of urinary bladder evaluation consists of examination in three planes. Due to this fact, the algorithm that consists of three parallel U-net algorithms is proposed. Each of the aforementioned U-nets is utilized in order to detect malignant mass from a CT image that represents a projection of the urinary bladder in one plane. A schematic representation of the proposed procedure is presented in Figure 5.

## 4. Transfer Learning Approach

The process of transfer learning can be mathematically defined by using a framework that consists of domain, task, and marginal probabilities definitions. If domain *D* is defined as a tuple of two elements [46]:feature space X, andmarginal probability P(X),
where *X* represents a sample data point. From the presented rules, a domain can be defined as:(1)D={X,P(X)},
where *X* is defined as:(2)$$X=\{x_{1},x_{2},\cdots ,x_{n}\},$$
where:(3)$$x_{i}\in X.$$

Furthermore, a task *T* can also be defined as a tuple that consists of the label space γ and objective function *O*. The presented objective function can be defined as:(4)O=P(γ|X).

If it is stated that source domain $D_{s}$ corresponds with source task $T_{s}$ and target domain $D_{t}$ corresponds with target task $T_{t}$, it can be stated that the objective of transfer learning process is to enable learning target conditional probability distribution $P(Y_{t}|X_{t})$ in $D_{t}$ by using knowledge gained from $D_{s}$ and $T_{s}$ where it is defined that:(5)$$D_{s}\neq D_{t},$$
and
(6)$$T_{s}\neq D_{s}.$$

For the purposes of this research, a transfer learning process can be described as utilization of pre-defined and pre-trained CNN architecture as a backbone to U-net. The aforementioned CNNs are pre-trained using one of the standard computer vision data sets.

For the purpose of this research, backbone CNNs were pre-trained using the ImageNet data set. Backbone CNNs represent the contractive part of a U-net, while the expansive part is added. As a contractive part of the U-net architecture, only the upper layers of the aforementioned pre-trained CNN architectures are used. On the other hand, the lower, fully connected layers of these CNN architectures are removed from the network in order to achieve the fully convolutional configuration required for achieving the semantic segmentation. During the training of the U-net, the layers in the contractive part of the network are frozen, and there is no change in their parameters.

Such a process is presented in Figure 6.

## 5. Used CNN Architectures

In this section, a brief overview of utilized CNN architectures will be provided. The first described CNN architecture, AlexNet, will be used only for plane recognition, while the other CNN architectures will be used only to design U-nets with pre-trained backbones.

### 5.1. Alexnet

In order to automatize the process of data set division according to planes, a CNN-based classification approach is proposed. For this purpose, AlexNet CNN architecture will be used. AlexNet represents one of the standard CNN architectures. It was developed by Alex Krizhevsky et al. and used to win the ImageNet competition [47]. AlexNet represents a deeper architecture that has started the trend of designing much deeper CNN architectures in recent years. For the purposes of this research, AlexNet is used only for plane recognition. The plane recognition problem represents a standard classification problem that can be solved by using less complex CNN architectures, such as AlexNet. AlexNet architecture has shown high classification performances when used for similar classification tasks in the biomedical field [12,48]. For these reasons, this architecture was chosen for the task of automatic CT image plane recognition.

A detailed description of the presented AlexNet CNN and all its layers is provided in Table 2.

### 5.2. Vgg-16

Another standard CNN architecture that will be used in this research is VGG-16. This architecture is also characterized with a deep architecture, even deeper than AlexNet. This architecture was developed in 2014 as an improvement of the AlexNet architecture [49]. It consists of a 16-layer architecture, from which the name VGG-16 was derived. Its main difference from the AlexNet architecture is smaller kernels in convolutional layers [50]. A difference with the AlexNet architecture is that this architecture will be used as a backbone of the U-net-based algorithm for the semantic segmentation. A detailed overview of the VGG-16 architecture is presented in Table 3.

### 5.3. Inception

Alongside more simple CNN architectures, for the design of pre-trained U-net backbones, more advanced CNN architectures are used as well. One of these architectures is Inception. The main difference between Inception and standard deep CNNs lays in the parallel configuration of an Inception block. Such an architecture is characterized by the parallel implementation of multiple convolution procedures with kernels of different sizes. All convolutions are performed on the same input feature map.

All outputs are concatenated and used as input for the next Inception layer. In this research, three different types of Inception modules will be used for the design of an Inception network. The first module used is based on dimension reduction, where larger kernels are replaced with successive convolution with smaller ones. A schematic representation of this Inception module is presented in Figure 7a. Furthermore, convolutions of size n×n can be replaced with equivalent consecutive combination of convolutions 1×n and n×1. Following the presented logic, it can be noticed that, for example, convolution 3×3 can be replaced with consecutive 1×3 and 3×1 convolutions. An illustration of the presented module is given in Figure 7b. The last inception block used to construct the Inception CNN used in this research is the configuration with parallel modules. A block scheme of such a configuration is presented in Figure 7c.

By using above presented Inception modules, the architecture presented in Table 4 is constructed, and this is used to construct the pre-trained backbone for the U-net semantic segmentation architecture.

### 5.4. Resnet

ResNet represents a more advanced CNN architecture that is based on the utilization of residual blocks. Residual block is constructed by using parallel Identity blocks in order to bypass convolutional layers. Such an approach is used in order to minimize the effect of vanishing gradients and to enable the construction of deeper CNN architectures [51]. A schematic representation of a residual block is presented in Figure 8.

For the purposes of this research, three different residual block-based CNN architectures are designed:ResNet50 [52],ResNet101 [53], andResNet152 [54].

### 5.5. Inception-Resnet

The last pre-defined CNN architecture used in this research is Inception-ResNet. Such an architecture represents a combination between Inception architecture and the approach of residual block utilization [55]. The presented approach is achieved by using Inception-residual blocks. A block scheme of such a block is presented in Figure 9.

## 6. Research Methodology

In order of determining the parallel U-net configuration with the highest segmentation performances, results achieved with standard and hybrid U-net architectures are compared. In this section, a brief description of semantic segmentation performance metrics is provided. Furthermore, the procedure of the U-net model selection procedure is described.

To maximize the segmentation performances of the proposed parallel U-net architecture, a grid search procedure is performed. Such a procedure is performed by changing U-net hyperparameters, re-training, and segmentation performance evaluation on the testing dataset. With this approach, the U-net configuration with the highest segmentation performances is included in the parallel algorithm. U-net hyperparameters used during the grid-search procedure are presented in Table 5.

### 6.1. Semantic Segmentation Performance Metrics

Comparison of designed U-nets is performed according to metrics for the semantic segmentation performance evaluation. As it is in the case of classification and regression, in this case, performances are also evaluated by using input and output data from the testing dataset. In this research, metrics:Intersection over union [56] andDice coefficient [57]
are used. Both metrics are based on a comparison of generated and true segmentation masks and represent the relationship between their shape and position. In the following paragraphs, a brief description of the aforementioned metrics will be provided.

#### 6.1.1. Intersection over Union

Intersection over Union (IoU) is a metric based on the ratio between the intersection of two segmentation maps and their union [58]. This ratio is defined as:(7)$$\begin{array}{c} IoU=\frac{X\cap Y}{X\cup Y}, \end{array}$$
where X∩Y represents an intersection and X∪Y represents a union. When the overlap of the actual and generated segmentation map is high, IoU will tend to
(8)IoU→1.

On the other hand, when the overlap is lower, IoU will tend to:(9)IoU→0.

From the presented extremes, it can be noticed that IoU, as a scalar measure for the semantic segmentation performance evaluation, will be part of the interval:(10)IoU∈[0,1].

#### 6.1.2. Dice Coefficient

Alongside IoU, the Dice coefficient (DSC) is also used as a metric for evaluation of semantic segmentation performances. DSC is defined as [59]:(11)$$\begin{array}{c} DSC=\frac{2|X\cap Y|}{\left|X|+|Y\right|}, \end{array}$$
where |X| represents the cardinality of the real, and |Y| represents the cardinality of the generated segmentation map. As it is in the case of IoU, when the overlap of the actual and generated segmentation map is high, DSC will tend to:(12)DSC→1.

When the overlap is low, DSC will tend to:(13)DSC→0.

From the presented extremes, it can be noticed that DSC will be a part of the interval:(14)DSC∈[0,1].

### 6.2. U-Net Model Selection

For purposes of selecting the best semantic segmentation model for each plane, a cross-validation procedure is introduced. The cross-validation procedure represents a standard procedure used in machine learning applications in order to define not only classification or semantic segmentation but also generalization performances. The aforementioned procedure is based on repeated re-training and testing of an ANN where data sets fractions (folds) that represent training and testing data sets are used interchangeably. The procedure is repeated until all folds are used for training and testing. The graphical representation of the described procedure is presented in Figure 10.

With the obtained information about CNNs classification or semantic segmentation performances in all cases, information about generalization performances can be derived. The average classification or semantic segmentation performances ($\bar{P}$) are defined as:(15)$$\bar{P}=\frac{1}{N}\sum_{i=1}^{N}P_{i},$$
where $P_{i}$ represents a result of the classification or semantic segmentation metrics obtained on a network trained and tested with data sets defined as case *i*. On the other hand, generalization performances (σ(P)) of a CNN are defined by using the standard deviation of $P_{i}$s achieved in all cases, or:(16)$$\sigma \left(P\right)=\sqrt{\frac{1}{N-1}\sum_{i=1}^{N}{\left(P_{i}-\bar{P}\right)}^{2}}$$
It has to be noted that CNNs with high classification and semantic segmentation performances will have:(17)$$P\to P_{max},$$
where
(18)$$P_{max}=1.$$

On the other hand, a CNN with high generalization performances will have:(19)σ(P)→0.

By using the presented criteria for performance evaluation, it can be noted that multi-objective criteria will be used for choosing the architecture that has the highest performances. All models are represented with tuples defined as:(20)$$T_{n}=\left\{\bar{DSC},\bar{IoU},\sigma \left(DSC\right),\sigma \left(Iou\right)\right\},$$
and are added to a set of tuples:(21)$$T=\{T_{1},T_{2},\ldots ,T_{N}\},$$

A set of tuples is sorted in such a manner that:(22)$$\pi_{1}\left(T_{q}\right)\leq \pi_{1}\left(T_{2}\right)\leq \cdots \leq \pi_{1}\left(T_{N}\right),$$
where $\pi_{1}$ represents the first element in a tuple, DSC. In the case when
(23)$$\pi_{1}\left(T_{n-1}\right)=\pi_{1}\left(T_{n}\right),$$
these two tuples are sorted that:(24)$$\pi_{3}\left(T_{n-1}\right)<\pi_{3}\left(T_{n}\right).$$

In this case, $\pi_{3}\left(T_{n}\right)$ can be defined as:(25)$$\pi_{3}\left(T_{n}\right)=\sigma \left(DSC\left(T_{n}\right)\right)$$

## 7. Results and Discussion

In this section, an overview of the achieved results is presented. In the first subsection, the results achieved with AlexNet CNN architecture for plane recognition are presented. In the second, third, and fourth subsections, the results achieved with U-net architectures are presented for each plane. At the end of the section, a brief discussion about the collected results is provided.

### 7.1. Plane Recognition

When the results of plane classification achieved by using AlexNet CNN architecture are observed, it can be noticed that the highest classification and generalization performances are achieved if the AlexNet architecture is trained by using RMS-prop optimizer for 10 consecutive epochs with data batches of 16. With the presented architecture $\bar{AUC_{micro}}$ value of 0.9999 and $\sigma (AUC_{micro})$ value of 0.0006 are achieved, as presented in Table 6.

If the change of classification and generalization performances through epochs are compared, it can be noticed that the highest results are achieved when AlexNet is trained for 10 consecutive epochs. Furthermore, it can be noticed that the performances are significantly lower when AlexNet is trained for higher number of epochs, pointing towards the over-fitting phenomena. If AlexNet is trained for just one epoch, the performances are also significantly lower as presented in Figure 11.

From the presented result, it can be concluded that AlexNet can be used for CT image plane recognition due the high classification and generalization performances.

### 7.2. Semantic Segmentation in Frontal Plane

If the results of semantic segmentation in the frontal plane are compared, it can be noticed that by applying the transfer learning paradigm, a significant improvement of semantic segmentation results is achieved. If a standard U-net architecture is utilized, $\bar{DSC}$ does not exceed 0.79 and $\bar{IoU}$ value does not exceed 0.77. Such results are pointing toward the conclusion that such a configuration has sufficient performance for practical application. Furthermore, if a transfer learning approach utilized a significant improvement of semantic segmentation and generalization performance is achieved.

The highest performances in both criteria are achieved if a pre-trained ResNet101 CNN architecture is used as a backbone for U-net. In this case, $\bar{DCS}$ values up to 0.9587 are achieved. Furthermore, $\bar{IoU}$ up to 0.9438 are achieved. When generalization performances are observed, it can be noticed that the lowest standard deviations are achieved when ResNet50 is used, followed by ResNet101. A detailed overview of the results achieved with and without pre-trained backbones is presented in Table 7, together with the hyper-parameters that achieved the highest results per backbone architecture.

If the change of semantic segmentation and generalization performances through epochs are observed for ResNet101, it can be noticed that the highest semantic segmentation and generalization performances are achieved when the U-net is trained for 125 consecutive epochs. When the network is trained for a higher number of epochs, it can be noticed that significantly poorer performances are achieved. Such a property can be attributed to the occurrence of the over-fitting phenomena. When the network is trained for a lower number of epochs, the results are also poorer, as presented in Figure 12.

### 7.3. Semantic Segmentation in Axial Plane

When the performances of semantic segmentation algorithms trained with CT images captured in the axial plane are compared, it can be noticed that by applying the transfer learning approach, significant improvement in both semantic segmentation and generalization performances is achieved. When the isolated learning paradigm is utilized, $\bar{DSC}$ and $\bar{IoU}$ are not exceeding 0.83 and 0.78, respectively. On the other hand, by utilization of a transfer learning approach, $\bar{DSC}$ and $\bar{IoU}$ values up to 0.9372 are achieved. From the detailed result presented in Table 8, it can be noticed that the highest semantic segmentation performances are achieved in a pre-trained ResNet50 architecture used as a backbone for the U-net.

When the change of $\bar{DCS}$ and σ(DSC) trough epochs is observed for the U-net designed with the pre-trained ResNet50 architecture as a backbone, it can be noticed that the highest semantic segmentation and generalization performances are achieved if the network is trained for 150 consecutive epochs. When the network is trained for a lower number of epochs, significantly lower performances could be noticed. Furthermore, if the network is trained for a higher number of epochs, the trend of decaying performances can be noticed, as presented in Figure 13. Such a property can be attributed to the over-fitting.

### 7.4. Semantic Segmentation in Sagittal Plane

The last data set used in this research is the data set of CT images captured in the sagittal plane. In this case, a significant improvement of semantic segmentation and generalization results can be observed if the transfer learning approach is utilized. In the case of standard U-net architectures, $\bar{DSC}$ and $\bar{IoU}$ do not exceed 0.86. On the other hand, if the transfer learning paradigm is utilized, significantly higher performances are achieved. By using this approach, $\bar{DSC}$ and $\bar{IoU}$ values up to 0.96660 are achieved, if a pre-trained VGG-16 architecture is used as a backbone for the U-net. Detailed results and models are presented in Table 9.

If the change of performances through epochs is observed, it can be noticed that the network with pre-trained VGG-16 architecture as a backbone achieved higher results if it is trained for a higher number of epochs. It is interesting to notice that the network has higher semantic segmentation performances if it is trained for 200 consecutive epochs. On the other hand, generalization performances are higher if the network is trained for 100 consecutive epochs, as presented in Figure 14.

### 7.5. Discussion

If $\bar{DCS}$ and σ(DSC) achieved on all three planes are compared, it can be noticed that the highest semantic segmentation performances are achieved on the sagittal plane, if pre-trained VGG-16 architecture is used as a backbone. For the case of the frontal and axial plane, slightly lover performances are achieved. On the other hand, if generalization performances are compared, it can be noticed that the highest σ(DSC) is achieved for the case of the sagittal plane. The best classification performances are achieved in the case of the frontal plane, as presented in Figure 15.

From the presented results, it can be seen that U-net for the case of the sagittal plane, although having the highest performance from the point of view of semantic segmentation, has significantly lower results from the point of view of generalization. Such a characteristic can be attributed to the fact that this part of the data set has a significantly lower number of images, in comparison with the other two parts. Such a lower number of images results in a lower number of training images in all five cases of the five-fold cross-validation procedure.

Such a lower number of training images results in lower semantic segmentation performances in some cross-validation cases and, thus, lower generalization performances. For these reasons, it can be concluded that, before the application of the proposed semantic segmentation system, more images captured in the sagittal plane need to be collected in order to increase the generalization performances of U-net for the semantic segmentation of images captured in the sagittal plane.

A similar trend can be noticed when the results measured with IoU are compared as presented in Figure 16. In this case, the difference between the results achieved on different planes, although lower, is still clearly visible. For these reasons, it can be concluded that the network used for images in the frontal plane has a more stable behavior. On the other hand, it can be concluded that the network used on images taken in the sagittal plane has much less stable behavior. This characteristic can clearly be attributed to the fact that the sagittal part of the data set has a significantly lower number of images.

The presented results are showing that there is a possibility for the application of the proposed system in clinical practice. It is shown that the high semantic segmentation performances enable the automatic evaluation of urinary bladder cancer spread. Furthermore, high generalization performances, especially in the case of the frontal and axial plane, indicate that the semantic segmentation system can be used for the evaluation of the new image data and new patients. The presented system can be used as an assistance system to medical professionals in order to improve clinical decision-making procedures.

## 8. Conclusions

According to the presented results, we concluded that the utilization of the transfer learning paradigm, in the form of pre-trained CNN architectures used as backbones for U-nets, can significantly improve the performances of semantic segmentation of urinary bladder cancer masses from CT images. Such an improvement can be noticed in both semantic segmentation and generalization performances. If the semantic segmentation performances are compared, $\bar{DSC}$ values of 0.9587, 0.9587, and 0.9660 are achieved for the case of the frontal, axial, and sagittal planes, respectively.

On the other hand, for the case of generalization performances, σ(DSC) of 0.0059, 0.0147, and 0.0486 are achieved for the case of the frontal, axial, and sagittal planes, respectively, suggesting the conclusion that the transfer learning approach opened the possibility for future utilization of such a system in clinical practice. Furthermore, U-nets for the semantic segmentation of urinary bladder cancer masses from images captured in the sagittal plane achieved significantly lower generalization performances. Such a characteristic can be assigned to the fact that the data set of sagittal images consists of the significantly lower number of images in comparison with two other data sets. According to the hypothesis questions, the conclusion can be summarized as:The design of a semantic segmentation system separately for each plane is possible.There is a possibility for the design of an automated system for plane recognition.By utilizing the transfer learning approach, significantly higher semantic segmentation and generalization performances are achieved.The highest performances are achieved if pre-trained ResNet101, ResNet50, and VGG-16 are used as U-net backbones for the semantic segmentation of images in the frontal, axial, and sagittal planes, respectively.

From the presented results, by utilizing the proposed approach, results in range of the results presented from the state-of-the-art approach were achieved. Furthermore, the proposed redundant approach increased the diagnostic performances and minimized the chance for incorrect diagnoses.

Future work will be based on improvements of the presented combination of classification and semantic segmentation algorithms by the inclusion of multiple classification algorithms before a parallel algorithm for the semantic segmentation. Furthermore, we plan to design a meta-heuristic algorithm for model selection with a fitness function that will be based on the presented multi-objective criteria. 

## Figures and Tables

**Figure 1 biology-10-01134-f001:**
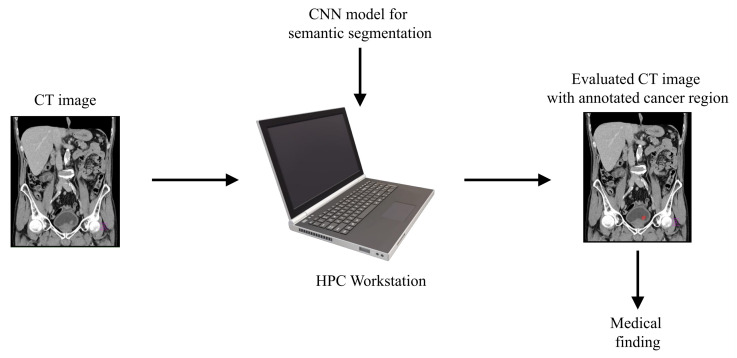
Dataflow diagram of the process of semantic segmentation of urinary bladder cancer from CT images.

**Figure 2 biology-10-01134-f002:**
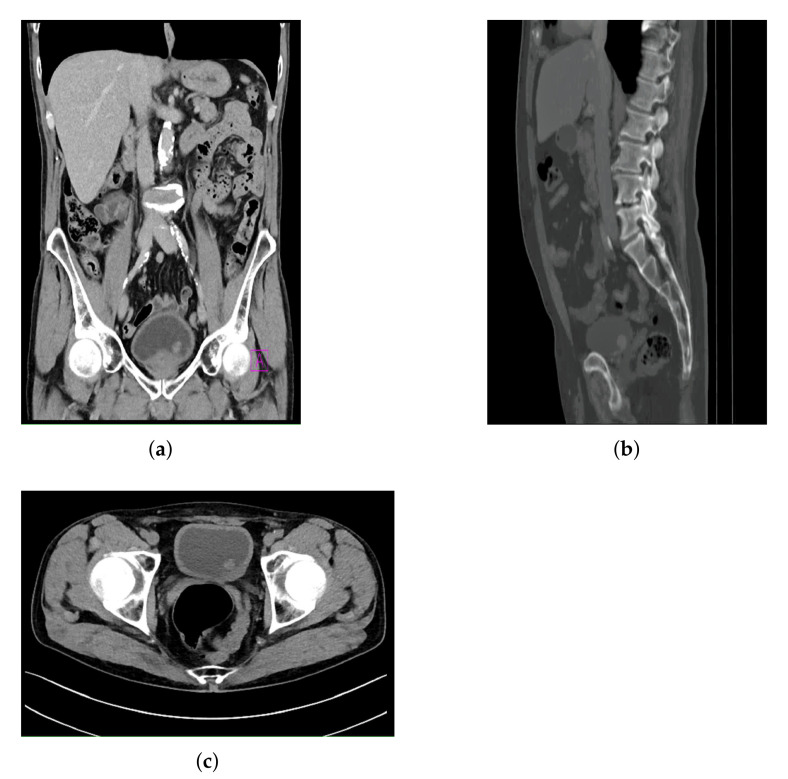
Examples of images contained in the dataset: (**a**) frontal plane; (**b**) sagittal plane; and (**c**) axial plane.

**Figure 3 biology-10-01134-f003:**
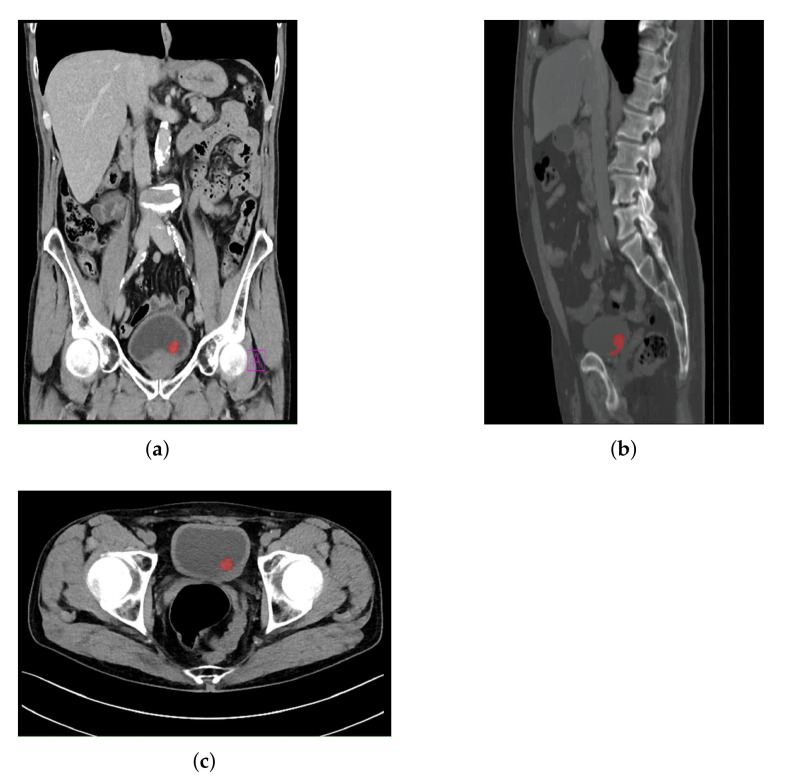
Examples of annotated images used for the creation of output masks: (**a**) frontal plane; (**b**) sagittal plane; and (**c**) axial plane.

**Figure 4 biology-10-01134-f004:**
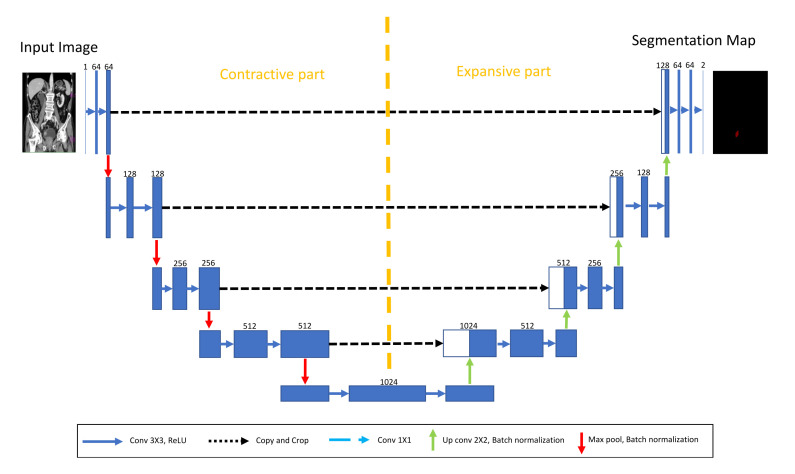
A block scheme of the proposed U-net architecture.

**Figure 5 biology-10-01134-f005:**
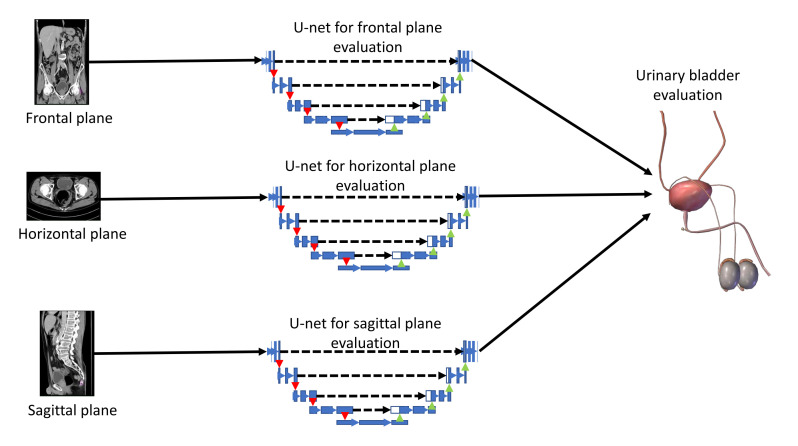
A block scheme of the proposed parallel U-net algorithm.

**Figure 6 biology-10-01134-f006:**
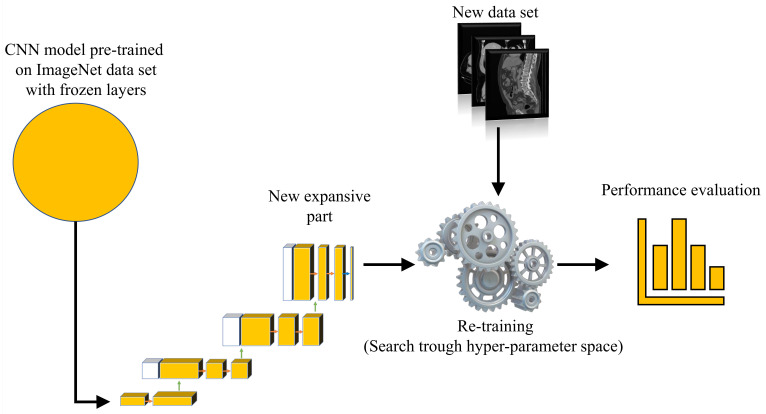
A block scheme of the proposed dataflow.

**Figure 7 biology-10-01134-f007:**
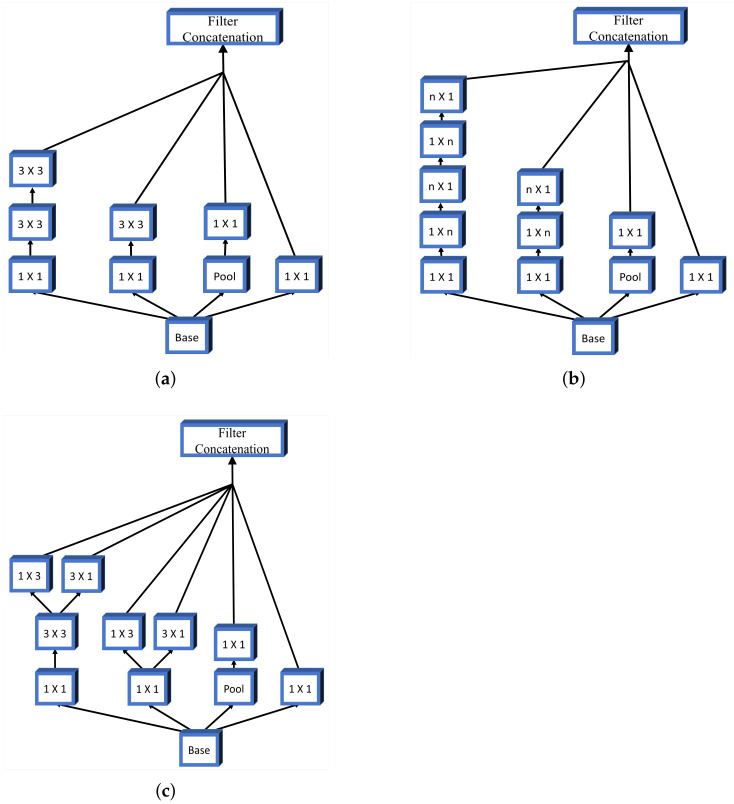
Block schemes of Inception modules (**a**) Inception-a; (**b**) Inception-b; and (**c**) Inception-c.

**Figure 8 biology-10-01134-f008:**
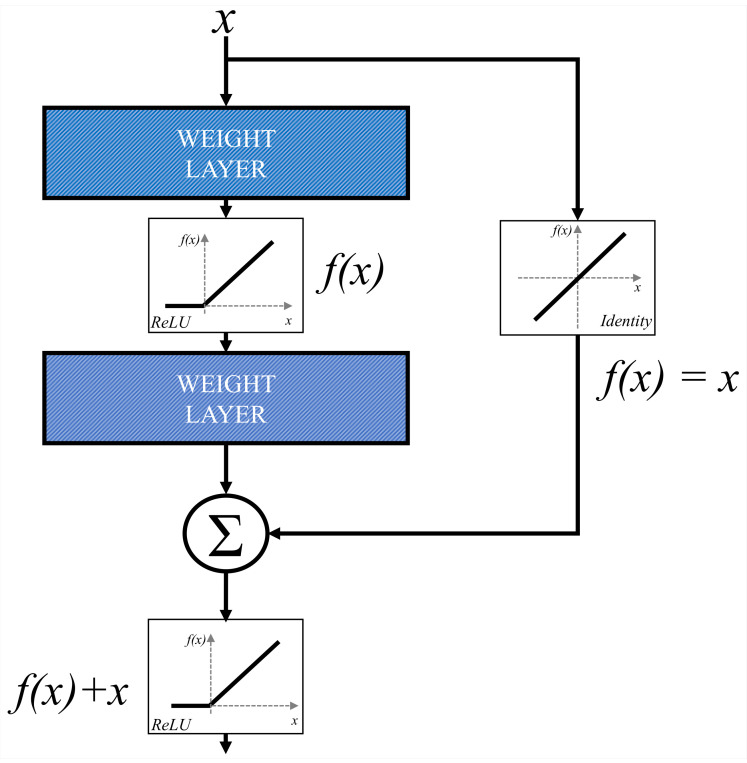
A block scheme of a residual block.

**Figure 9 biology-10-01134-f009:**
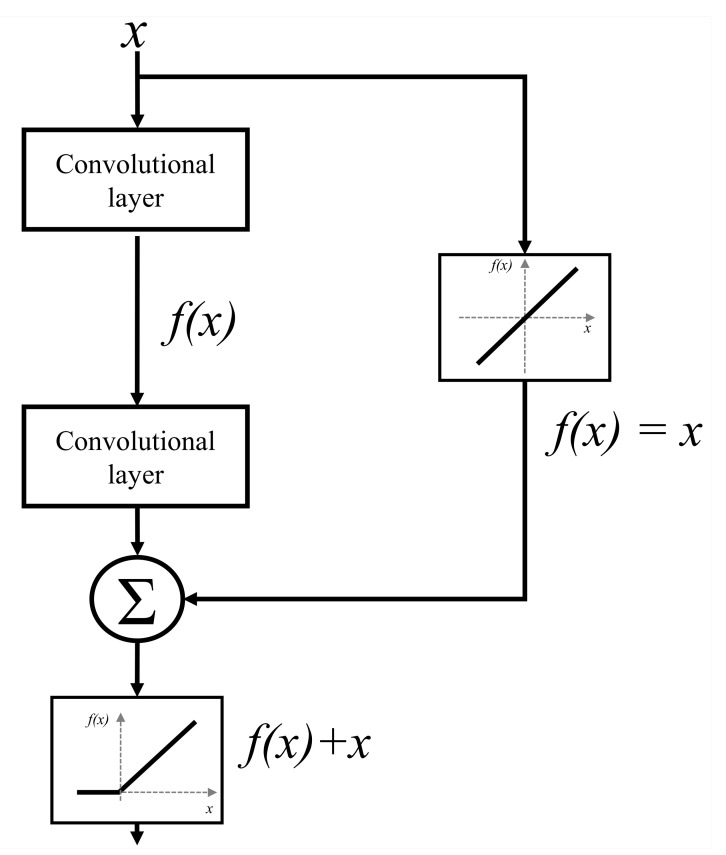
A block scheme of an Inception-residual block.

**Figure 10 biology-10-01134-f010:**
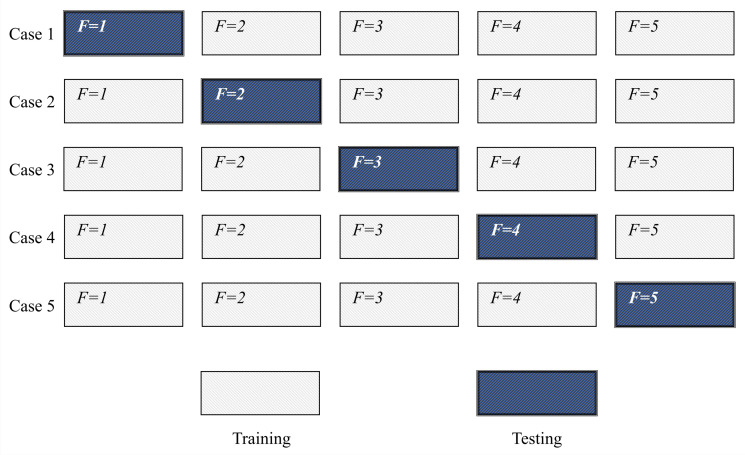
A schematic representation of the five-fold cross-validation procedure.

**Figure 11 biology-10-01134-f011:**
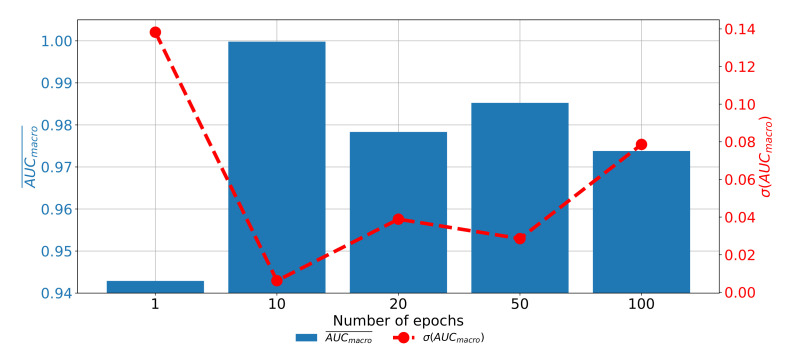
The change of $\bar{AUC_{micro}}$ and $\sigma (AUC_{micro})$ through the number of epochs achieved with AlexNet for plane recognition.

**Figure 12 biology-10-01134-f012:**
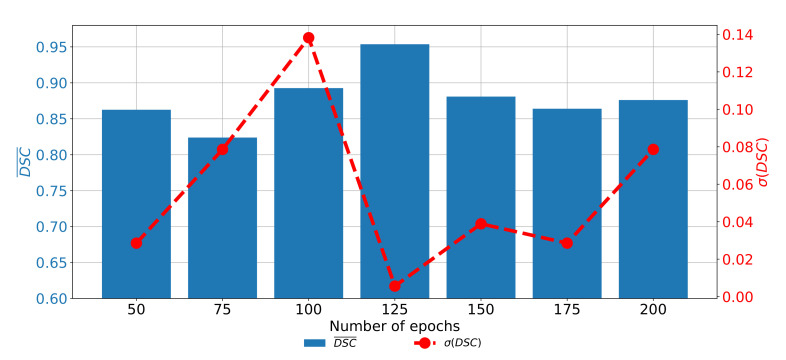
The change of $\bar{DCS}$ and σ(DSC) through a number of epochs achieved with pre-trained ResNet101 as U-net backbone for the semantic segmentation of urinary bladder cancer masses from CT images in frontal plane.

**Figure 13 biology-10-01134-f013:**
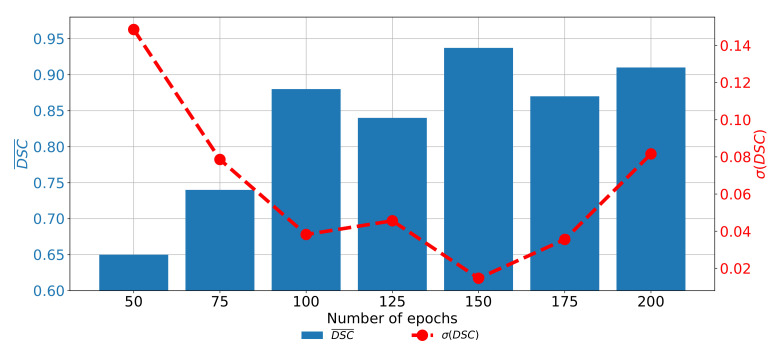
The change of $\bar{DCS}$ and σ(DSC) through a number of epochs achieved with a pre-trained ResNet50 architecture as U-net backbone for the semantic segmentation of urinary bladder cancer masses from CT images in axial plane.

**Figure 14 biology-10-01134-f014:**
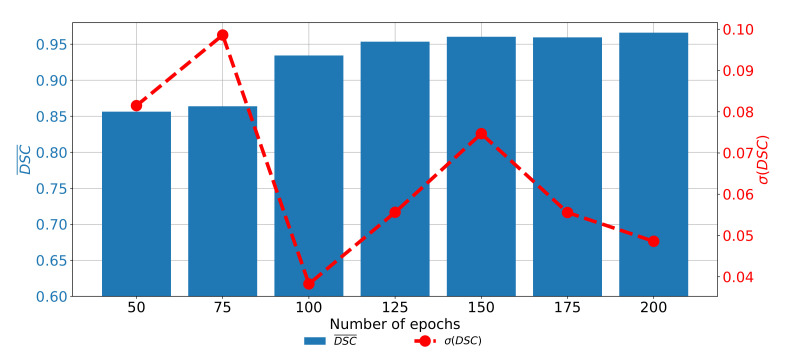
The change of $\bar{DCS}$ and σ(DSC) through a number of epochs achieved with pre-trained VGG-16 architecture as U-net backbone for the semantic segmentation of urinary bladder cancer masses from CT images in the sagittal plane.

**Figure 15 biology-10-01134-f015:**
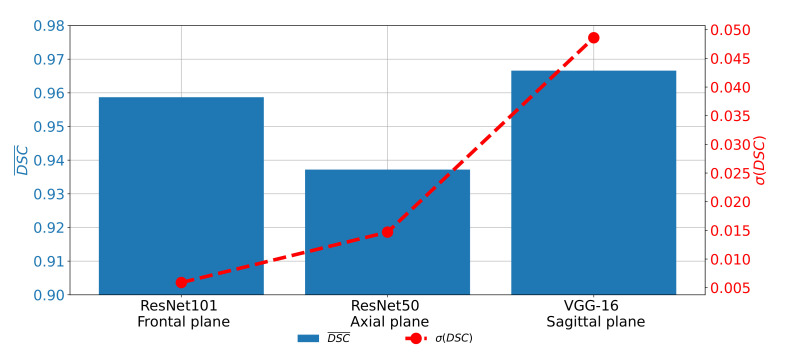
Comparison of $\bar{DCS}$ and σ(DSC) achieved with the most successful architectures for each plane.

**Figure 16 biology-10-01134-f016:**
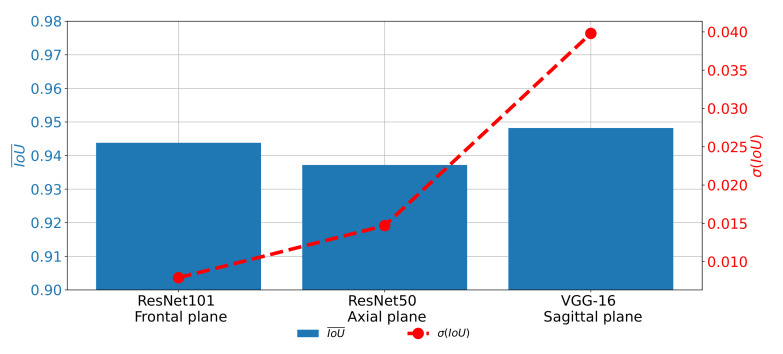
Comparison of $\bar{IoU}$ and σ(IoU) achieved with the most successful architectures for each plane.

**Table 1 biology-10-01134-t001:** Original data set distribution.

Plane	Number of Images
Frontal	4413
Axial	4993
Sagittal	996

**Table 2 biology-10-01134-t002:** Description of AlexNet architecture (C—convolutional layer, P—Max pooling, and FC—fully connected).

Layer	Type	Feature Map	Size	Kernel Size	Stride	Activation Function
Input	Image	1	227×227×1	-	-	-
1	C	96	55×55×96	11×11	4	ReLU
	P	96	27×27×96	3×3	2	-
2	C	256	27×27×256	5×5	1	ReLU
	P	256	13×13×256	3×3	2	-
3	C	384	13×13×384	3×3	1	ReLU
4	C	384	13×13×384	3×3	1	ReLU
5	C	256	13×13×256	3×3	1	ReLU
	P	256	6×6×256	3×3	2	-
6	FC	-	9216	-	-	ReLU
7	FC	-	4096	-	-	ReLU
8	FC	-	4096	-	-	ReLU
Output	FC	-	4	-	-	Softmax

**Table 3 biology-10-01134-t003:** Description of VGG-16 architecture (C—convolutional layer, P—Max pooling, and FC—fully connected).

Layer	Type	Activation Function
Input	Image	-
1	2 X C	ReLU
	P	-
3	2 X C	ReLU
	P	-
5	3 X C	ReLU
	P	-
8	3 X C	ReLU
	P	-
11	3 X C	ReLU
	P	-
14	FC	ReLU
15	FC	ReLU
16	FC	ReLU
Output	FC	Softmax

**Table 4 biology-10-01134-t004:** Layer configuration of an InceptionV3 architecture.

Layer Type	Patch Size/Stride/Remark	Input Size
Convolutional	3 × 3/2	299 × 299 × 3
Convolutional	3 × 3/1	149 × 149 × 32
Convolutional + Padding	3 × 3/1	147 × 147 × 32
Pooling	3 × 3/2	147 × 147 × 64
Convolutional	3 × 3/1	73 × 73 × 64
Convolutional	3 × 3/2	71 × 71 × 80
Convolutional	3 × 3/1	35 × 35 × 288
3 × Inception-a	As in Figure 7a	35 × 35 × 288
5 × Inception-b	As in Figure 7b	17 × 17 × 768
2 × Inception-c	As in Figure 7c	8 × 8 × 1280
Pooling	8 × 8	8 × 8 × 2048
Linear	Logits	1 × 1 × 2048
Softmax	Classification	1 × 1 × 1000

**Table 5 biology-10-01134-t005:** Overview of U-net hyperparameters used during grid-search procedure.

Solver	Batch Size	Number of Epochs
Adam	1	50
AdaMax	2	75
Adagrad	4	100
AdaDelta	8	125
RMSprop	16	150
Nadam	-	175
-	-	200

**Table 6 biology-10-01134-t006:** The AlexNet architecture with the highest plane recognition performances.

Solver	Epochs	Batch Size	$\bar{{\mathrm{AUC}}_{\mathrm{micro}}}$	$\sigma ({\mathrm{AUC}}_{\mathrm{micro}})$
RMS-prop	10	16	0.9999	0.0006

**Table 7 biology-10-01134-t007:** Results achieved with images in the frontal plane.

Backbone Architecture	Solver	Epochs	Batch	$\bar{\mathrm{DSC}}$	$\bar{\mathrm{IoU}}$	σ(DSC)	σ(IoU)
None	AdaMax	25	2	0.7846	0.7655	0.0439	0.0444
VGG-16	Nadam	150	4	0.9134	0.9011	0.0816	0.0787
InceptionV3	RMS-prop	75	2	0.9031	0.8955	0.0149	0.0147
ResNet50	Adam	50	8	0.9314	0.9258	0.0019	0.0019
ResNet101	Nadam	50	2	0.9587	0.9438	0.0059	0.0079
ResNet152	RMS-prop	100	8	0.8121	0.8067	0.0082	0.0092
Inception-ResNet	Nadam	175	2	0.8991	0.8962	0.1212	0.1209

**Table 8 biology-10-01134-t008:** Results achieved with images in the axial plane.

Backbone Architecture	Solver	Epochs	Batch	$\bar{\mathrm{DSC}}$	$\bar{\mathrm{IoU}}$	σ(DSC)	σ(IoU)
None	AdaMax	50	4	0.8347	0.7832	0.0711	0.0948
VGG-16	Adam	150	2	0.8804	0.8656	0.2456	0.2432
InceptionV3	RMS-prop	150	4	0.9147	0.9147	0.0051	0.0051
ResNet50	Adam	150	4	0.9372	0.9372	0.0147	0.0147
ResNet101	Nadam	75	8	0.9069	0.9069	0.0203	0.0203
ResNet152	RMS-prop	100	4	0.8549	0.8421	0.0563	0.0671
Inception-ResNet	Adam	100	8	0.8456	0.8362	0.0514	0.0548

**Table 9 biology-10-01134-t009:** Results achieved with images in the sagittal plane.

Backbone Architecture	Solver	Epochs	Batch	$\bar{\mathrm{DSC}}$	$\bar{\mathrm{IoU}}$	σ(DSC)	σ(IoU)
None	Adam	10	4	0.8639	0.7938	0.0845	0.0917
VGG-16	Adam	200	2	0.9660	0.9482	0.0486	0.0398
InceptionV3	RMS-prop	75	8	0.8754	0.8497	0.0654	0.0758
ResNet50	AdaMax	150	4	0.8448	0.8358	0.0256	0.0262
ResNet101	AdaMax	75	2	0.8356	0.8280	0.0129	0.0134
ResNet152	Adam	100	2	0.8726	0.8655	0.0844	0.7753
Inception-ResNet	Adam	200	2	0.8454	0.8385	0.0275	0.0288

## Data Availability

The data presented in this study are available on request from the corresponding author, if data sharing is approved by ethics committee. The data are not publicly available due to data protection laws and conditions stated by the ethics committee.

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
