# Peer review of "Semantic Segmentation of Urinary Bladder Cancer Masses from CT Images: A Transfer Learning Approach"

_biology, 2021, doi:10.3390/biology10111134_

Round 1

Reviewer 1 Report

The authors present a novel approach to the application of U-net architectures for semantic segmentation of bladder cancer. A parallel system based on three different CT imaging planes and the application of CNN for plane classification was proposed. In addition, the authors suggest the application of the transfer learning methodology. The paper is written correctly and the scientific contribution is quite clear. However, in order for the paper to be considered suitable for the publication, several comments need to be addressed.

#1 The graphical overview should be described in more detail. Please specify what is expected as input and what as system output according to the diagram shown in the figure.

#2 In the caption for Figure 2, a red area is mentioned. However, there is no red area in any of the images. Please review this caption.

#3 Lines 207-212 state that pre-trained CNN architectures trained on the ImageNet dataset are used for transfer learning. In order to improve understanding, it is necessary to clarify how CNN architectures were used as backbones. What about layer freezing?

#4 Why do the authors think the AlexNet architecture is sufficient to recognize a plane? Please clarify in 5.1.

 #5 In Discussion, please discuss the application of the subject system in clinical practice. How do the results affect the applicability of this system?

Author Response

The authors want to thank Reviewer 1 for constructive comments that have significantly increased the paper’s quality. The authors sincerely hope that the manuscript with performed changes will be suitable for publication in Biology. The changes in the manuscript performed according to the comments of Reviewer 1 are marked with blue color.

The authors present a novel approach to the application of U-net architectures for semantic segmentation of bladder cancer. A parallel system based on three different CT imaging planes and the application of CNN for plane classification was proposed. In addition, the authors suggest the application of the transfer learning methodology. The paper is written correctly and the scientific contribution is quite clear. However, in order for the paper to be considered suitable for the publication, several comments need to be addressed.

#1 The graphical overview should be described in more detail. Please specify what is expected as input and what as system output according to the diagram shown in the figure.

The description of the graphical overview is expanded and following text is added:

“As input data to the semantic segmentation system, images of lower abdomen collected with CT are used. An input image, captured in three planes. First, the classification is performed, in order to determine the plane in which the image was captured. After the plane is determined, the image is used as an input to U-net architecture for that particular plane. Trained U-net architecture is used to create the output mask. Output mask represents the region of an image where urinary bladder cancer is present. Such an output enables automated evaluation of urinary bladder that results with an annotated image that is used to determine the urinary bladder cancer spread.”

#2 In the caption for Figure 2, a red area is mentioned. However, there is no red area in any of the images. Please review this caption.

The caption has been revised, and was changed to:

“Examples of images contained in the dataset: (a) frontal plane; (b) sagittal plane; (c) axial plane”

#3 Lines 207-212 state that pre-trained CNN architectures trained on the ImageNet dataset are used for transfer learning. In order to improve understanding, it is necessary to clarify how CNN architectures were used as backbones. What about layer freezing?

According to this comment, this part of the manuscript is expanded. In the description of the transfer learning approach, the following text is added:

“As a contractive part of the U-net architecture, only upper layers of the aforementioned pre-trained CNN architectures are used. On the other hand, lower, fully connected layers of these CNN architectures are removed from the network in order to achieve the fully convolutional configuration, required for achieving the semantic segmentation. During the training of the U-net, the layers in the contractive part of the network are frozen and there is no change in their parameters.”

#4 Why do the authors think the AlexNet architecture is sufficient to recognize a plane? Please clarify in 5.1.

According to this comment, an explanation why AlexNet was chosen for plane recognition is added:

“The plane recognition problem represents a standard classification problem that can be solved by using less complex CNN architectures, such as AlexNet. AlexNet architecture has shown high classification performances while used for similar classification tasks in the biomedical field []. For these reasons, this architecture was chosen for the task of automatic CT image plane recognition.”

Alongside stated above, the following text is added in 7.1.:

“From the presented result, it can be concluded that AlexNet can be used for CT image plane recognition due high classification and generalization performances.”

 #5 In Discussion, please discuss the application of the subject system in clinical practice. How do the results affect the applicability of this system?

The following text is added in the Discussion:

“The presented results are showing that there is a possibility for the application of the proposed system in clinical practice. It is shown that the high semantic segmentation performances are enabling automatic evaluation of urinary bladder cancer spread. Furthermore, high generalization performances, especially in the case of the frontal and axial plane, are pointing towards the fact that the semantic segmentation system can be used for the evaluation of the new image data and new patients. The presented system can be used as an assistance system to medical professionals in order to improve clinical decision-making procedure.”

Reviewer 2 Report

In the manuscript “Semantic Segmentation of Urinary Bladder Cancer Masses From CT Images: a Transfer Learning Approach” by Sandi Baressi Šegota, Ivan Lorencin, Klara Smolić, Nikola Andelić, Dean Markić, Vedran Mrzljak, Daniel Štifanić, Jelena Musulin, Josip Španjol and Zlatan Car, a novel approach to the semantic segmentation of urinary bladder cancer masses from CT images is proposed. I recommend that the manuscript should be published, but I have some minor issues with the manuscript that need to be resolved before acceptance.

  1. I) State of the art should be expanded by adding the following articles:

  1. Monteiro, M., Newcombe, V. F., Mathieu, F., Adatia, K., Kamnitsas, K., Ferrante, E., ... & Glocker, B. (2020). Multiclass semantic segmentation and quantification of traumatic brain injury lesions on head CT using deep learning: an algorithm development and multicentre validation study. The Lancet Digital Health, 2(6), e314-e322.
  2. Anthimopoulos, M., Christodoulidis, S., Ebner, L., Geiser, T., Christe, A., & Mougiakakou, S. (2018). Semantic segmentation of pathological lung tissue with dilated fully convolutional networks. IEEE journal of biomedical and health informatics, 23(2), 714-722.
  3. Meraj, T., Rauf, H. T., Zahoor, S., Hassan, A., Lali, M. I., Ali, L., ... & Shoaib, U. (2021). Lung nodules detection using semantic segmentation and classification with optimal features. Neural Computing and Applications, 33(17), 10737-10750.
  4. Koitka, S., Kroll, L., Malamutmann, E., Oezcelik, A., & Nensa, F. (2021). Fully automated body composition analysis in routine CT imaging using 3D semantic segmentation convolutional neural networks. European radiology, 31(4), 1795-1804.

II) In the caption for Figure 2 it is stated that the red area represents a bladder wall thickening, but there is no red area on the images whatsoever. Please rewrite this caption correctly.

III)  I recommend that all tables in the paper should be redesigned to match with style to Tables 6, 7, 8 and 9

IV) In Conclusion a comparison of the achieved results is given, but without exact numerical values. I recommend repeating the best results from the previous section to increase the paper's readability.

Author Response

The authors want to thank Reviewer for constructive comments that have significantly increased the paper’s quality. The authors sincerely hope that the manuscript with performed changes will be suitable for publication in Biology. The changes in the manuscript performed according to the comments of Reviewer are marked with red color.

In the manuscript “Semantic Segmentation of Urinary Bladder Cancer Masses From CT Images: a Transfer Learning Approach” by Sandi Baressi Šegota, Ivan Lorencin, Klara Smolić, Nikola Andelić, Dean Markić, Vedran Mrzljak, Daniel Štifanić, Jelena Musulin, Josip Španjol and Zlatan Car, a novel approach to the semantic segmentation of urinary bladder cancer masses from CT images is proposed. I recommend that the manuscript should be published, but I have some minor issues with the manuscript that need to be resolved before acceptance.

  1. I) State of the art should be expanded by adding the following articles: 
  • Monteiro, M., Newcombe, V. F., Mathieu, F., Adatia, K., Kamnitsas, K., Ferrante, E., ... & Glocker, B. (2020). Multiclass semantic segmentation and quantification of traumatic brain injury lesions on head CT using deep learning: an algorithm development and multicentre validation study. The Lancet Digital Health, 2(6), e314-e322.
  • Anthimopoulos, M., Christodoulidis, S., Ebner, L., Geiser, T., Christe, A., & Mougiakakou, S. (2018). Semantic segmentation of pathological lung tissue with dilated fully convolutional networks. IEEE journal of biomedical and health informatics, 23(2), 714-722.
  • Meraj, T., Rauf, H. T., Zahoor, S., Hassan, A., Lali, M. I., Ali, L., ... & Shoaib, U. (2021). Lung nodules detection using semantic segmentation and classification with optimal features. Neural Computing and Applications, 33(17), 10737-10750.
  • Koitka, S., Kroll, L., Malamutmann, E., Oezcelik, A., & Nensa, F. (2021). Fully automated body composition analysis in routine CT imaging using 3D semantic segmentation convolutional neural networks. European radiology, 31(4), 1795-1804.

The following text was added, including the above citations in the state-of-the-art section:

“Monteiro et al. (2020) [] demonstrate the use of CNNs for multiclass semantic segmentation and quantification of traumatic brain injury lesions on head CT images. The patient data is collected in the period between 2014 and 2017, on which the CNN was trained for the task of voxel-level multiclass segmentation/classification. The authors find that such an approach shows a high-quality volumetric lesion estimate and may potentially be applied for personalized treatment strategies and clinical research. Anthimopoulos et al. (2018) [] demonstrate the use of Dilated Fully Convolutional Networks for the task of semantic segmentation on pathological lung tissue. The authors use 172 sparsely annotated CT scans within a cross-validation training scheme, with training done in a semi-supervised mode, using labeled and unlabeled image regions. The results show that the proposed methodology achieves significant performance improvement in comparison to previous research in the field. Another research regarding segmentation on lung CTs is performed by Meraj et al. (2021) [], with the goal of performing lung nodules detection. Authors use a publicly available dataset, Lung Image Database Consortium, upon which filtering and noise removal are applied. Authors use adaptive thresholding and semantic segmentation for unhealthy lung nodule detection, with feature extraction performed via principal component extraction. Such an approach shows the results of 99.23\% accuracy when logit boost classifier is applied. Koitka et al. (2021) [] propose an automatic manner of body composition analysis, with the goal of application during routine CT scanning. Authors utilize 3D semantic segmentation CNNs, applying them on a dataset consisting of 50 CTs annotated on every fifth axial slice split into 80:20 ratio. Authors achieve high results, with the average dice scores reaching 0.9553, indicating a successful application of CNNs for the purpose of body composition determination.”

  1. II) In the caption for Figure 2 it is stated that the red area represents a bladder wall thickening, but there is no red area on the images whatsoever. Please rewrite this caption correctly.

The caption has been revised, and was changed to:

“Examples of images contained in the dataset: (a) frontal plane; (b) sagittal plane; (c) axial plane”

III) I recommend that all tables in the paper should be redesigned to match with style to Tables 6, 7, 8 and 9

Tables 1 through 5 have been restyled to use the same design as Tables 6 through 9.

  1. IV) In Conclusion a comparison of the achieved results is given, but without exact numerical values. I recommend repeating the best results from the previous section to increase the paper's readability.

According to this comment, the following text was added:

“If the semantic segmentation performances are compared, it can be noticed that DSC values of 0.9587, 0.9587, and 0.9660 are achieved for the case of the frontal, axial, and sagittal plane, respectively. On the other hand, for the case of generalization performances, it can be noticed that σ(DSC) of 0.0059, 0.0147, and 0.0486 are achieved for the case of the frontal, axial, and sagittal plane, respectively.”

Reviewer 3 Report

This manuscript use transfer learning approach to the semantic segmentation of urinary bladder cancer masses from CT images to increase clinical decision making and diagnostics for bladder cancer.

This is an interesting article to improve clinical diagnosis accuracy and decrease unnecessary procedure.

Author Response

The authors want to thank the Reviewer for constructive comments that have significantly increased the paper’s quality. The authors sincerely hope that the manuscript with performed changes will be suitable for publication in Biology. As requested, the manuscript was checked and minor spellcheck and grammar errors have been corrected throughout.

Reviewer 4 Report

I have the following comments:

-Please briefly emphasize the conclusions of the study and their potential relevance to clinical practice at the end of the abstract.

-In the Introduction, the description of the CT methodology used to provide high resolution images of the urinary bladder (from which stacks of multiplanar reformatted images can be obtained) should include more detailed information, regarding e.g. currently available multidetector CT technology, iodinated contrast medium administration, basic post-processing techniques of CT images, etc. A radiologist could ideally be asked for expert advice on this.

-Table 1. Why were sagittal images (996) substantially less than frontal (4413) and "horizontal" (4993 - please use "axial" instead of "horizontal" throughout the entire manuscript)? Could this imbalance affect the study findings?

Author Response

The authors want to thank Reviewer for constructive comments that have significantly increased the paper’s quality. The authors sincerely hope that the manuscript with performed changes will be suitable for publication in Biology. The changes in the manuscript performed according to the comments of Reviewer are marked with green color.

I have the following comments:

-Please briefly emphasize the conclusions of the study and their potential relevance to clinical practice at the end of the abstract.

According to this comment, the following sentences are added to the end of the Abstract:

“From the listed results, it can be noticed that the proposed semantic segmentation system works with high performances, both from semantic segmentation and generalization standpoints. The presented results are pointing towards the conclusion that there is a possibility for the utilization of the semantic segmentation system in clinical practice.”

-In the Introduction, the description of the CT methodology used to provide high resolution images of the urinary bladder (from which stacks of multiplanar reformatted images can be obtained) should include more detailed information, regarding e.g. currently available multidetector CT technology, iodinated contrast medium administration, basic post-processing techniques of CT images, etc. A radiologist could ideally be asked for expert advice on this.

According to this comment, the following text is added:

"Computed tomography (CT) scans are a commonly used medical diagnostic imaging method in which multiple X-ray measurements are taken to produce tomographic 53 images of a patient’s body, allowing the examination of patient’s organs without a need for a more invasive procedure [13]. Today, multi-detector CT scanners, having 64 to 320 rows of detectors, are being used combined with helical image acquisition techniques, as they minimize the exposure to the radiation and can generate sagittal, axial, and frontal images of the body during a single breath-hold [14]. CT urography and CT of the abdomen and pelvis are contrast-enhanced imaging methods for detection and staging of bladder cancer that are able to differentiate healthy from the cancer affected regions of the bladder [15,16]. Non-ionic monomer iodinated contrast agents are administered intravenously for arterial opacification and parenchymal enhancement which help better delineation of soft tissue [17]. Acquired images are regularly inspected and interpreted by the radiologist with detailed descriptions of the bladder may have been affected [18]. In post-processing radiologist can mark and measure the suspected tumor or create a 3D recreation of a urinary system. Detection of urinary bladder cancer by using CT urography has shown high performances, and it can be concluded that CT urography can be used alongside cystoscopy for detecting urinary bladder cancer [19]."

-Table 1. Why were sagittal images (996) substantially less than frontal (4413) and "horizontal"

According to the comment, the following explanation is added:

“When the numbers of images in each plane are compared, it can be seen that a significantly lower number of images are captured for the case of the sagittal plane, in comparison with the frontal and axial plane. Such an imbalance is a consequence of the fact that the CT urography procedures are, in the case of this research, dominantly performed in frontal and axial planes only. Furthermore, images captured in the sagittal plane are captured with lower density, resulting in a lower number of images per patient."

 (4993 - please use "axial" instead of "horizontal" throughout the entire manuscript)? 

Horizontal is replaced with axial throughout the entire manuscript.

Could this imbalance affect the study findings?

In the original version of the manuscript, the influence of the lower number of images captured in the sagittal plane on generalization performances of U-net for semantic segmentation of images in the sagittal plane was addressed in the Discussion. However, in order to emphasize these observations, the following text was added:

“Such a lower number of images results in a lower number of training images in all 5 cases of the 5-fold cross-validation procedure. Such a lower number of training images results in lower semantic segmentation performances in some cross-validation cases, thus lower generalization performances. For these reasons, it can be concluded that before the application of the proposed semantic segmentation system, more images captured in the sagittal plane need to be collected in order to increase the generalization performances of U-net for semantic segmentation of images captured in the sagittal plane.”